

# Federated learning for digital twin applications: a privacy-preserving and low-latency approach

Jie Li[1] and Dong Wang[2]

[1] School of Management, Zhejiang University of Technology, Hangzhou, China
[2] College of Computer Science and Technology, Zhejiang University of Technology, Hangzhou, China

## ABSTRACT

The digital twin (DT) concept has recently gained widespread application for mapping the state of physical entities, enabling real-time analysis, prediction, and optimization, thereby enhancing the management and control of physical systems. However, when sensitive information is extracted from physical entities, it faces potential leakage risks, as DT service providers are typically honest yet curious. Federated learning (FL) offers a new distributed learning paradigm that protects privacy by transmitting model updates from edge servers to local devices, allowing training on local datasets. Nevertheless, the training parameters communicated between local mobile devices and edge servers may contain raw data that malicious adversaries could exploit. Furthermore, variations in mapping bias across local devices and the presence of malicious clients can degrade FL training accuracy. To address these security and privacy threats, this paper proposes the FL-FedDT scheme—a privacy-preserving and low-latency FL method that employs an enhanced Paillier homomorphic encryption algorithm to safeguard the privacy of local device parameters without transmitting data to the server. Our approach introduces an improved Paillier encryption method with a new hyperparameter and pre-calculates multiple random intermediate values during the key generation stage, significantly reducing encryption time and thereby expediting model training. Additionally, we implement a trusted FL global aggregation method that incorporates learning quality and interaction records to identify and mitigate malicious updates, dynamically adjusting weights to counteract the threat of malicious clients. To evaluate the efficiency of our proposed scheme, we conducted extensive experiments, with results validating that our approach achieves training accuracy and security on par with baseline methods, while substantially reducing FL iteration time. This enhancement contributes to improved DT mapping and service quality for physical entities. (The code for this study is publicly available on GitHub at: https://github.com/fujianU/federated-learning. The URL address of the MNIST dataset is: https://gitcode.com/Resource-Bundle-Collection/d47b0/overview?utm_source=pan_gitcode&index=top&type=href&;.)

Corresponding author
Dong Wang,
122224120039@zjut.edu.cn

## INTRODUCTION

The ongoing deployment of 5th generation mobile communication technology (5G) and the emergence of 6th generation mobile communication technology (6G) have garnered considerable attention from both industry and academia, with digital twin (DT) technology becoming a key area of focus (*Wang et al., 2023b*; *Tang et al., 2024*; *Abd Wahab et al., 2024*). DT is utilized to create real-time mappings and models of physical entities in heterogeneous environments, thereby enabling real-time analysis, learning, prediction, and interaction with perception-driven data from the physical world (*Singh et al., 2021*). At its core, DT bridges the physical and virtual realms by receiving data and replicating physical behaviors. However, the distributed nature of data in real-world applications introduces significant challenges concerning data security and privacy, which hinder the widespread adoption of DT. This is primarily due to the transmission of plaintext data between servers and local devices (LDs), making it vulnerable to interception by third parties once it leaves the LD. While most servers providing DT synchronization services are generally trusted, they may still be tempted to access sensitive information (*Wang et al., 2023c*). Additionally, the frequent use of wireless communication for DT synchronization increases the risk of malicious actors intercepting transmitted data *via* open channels. Even if only partial data is obtained, complete information may be reconstructed through various attack strategies, such as chosen-plaintext attacks (*Alcaraz & Lopez, 2022*).

Several studies (*Stergiou, Bompoli & Psannis, 2023*; *Son et al., 2022*; *He et al., 2023*) have already addressed the security and privacy concerns of DT in heterogeneous environments. In *Stergiou, Bompoli & Psannis (2023)*, the authors proposed a secure communication scheme for a sustainable cloud system supporting DT. This scheme implements computational optimizations on traditional encryption algorithms, including Advanced Encryption Standard (AES), RC5, and Rivest–Shamir–Adleman (RSA), thereby enhancing system efficiency and security. In *Son et al. (2022)*, the authors introduced a system model for securely sharing DT data. The proposed model leverages cloud computing to facilitate efficient data sharing while integrating blockchain technology to ensure data verifiability and integrity. In *He et al. (2023)*, the authors investigated the security and privacy aspects of vehicular DT. By analyzing the network architecture of vehicular DT, they identified potential threats and open research challenges from a security and privacy perspective. Additionally, the study proposed feasible authentication mechanisms to enhance the security of vehicular DT systems. The primary approach involves collecting data from terminal devices and then processing the aggregated data on edge servers. However, in such a model, sensitive information stored on the device is beyond the control of the end user, which increases the risk of unauthorized disclosure. Moreover, the advanced capabilities of mobile devices, driven by 6G communication technologies, will facilitate the upload of large volumes of locally generated data to remote servers. Consequently, the frequent exchange of data between edge servers and devices results in significant energy consumption and an increase in transmission delays. As a result, the processing of data directly on LDs, rather than frequent transmission to centralized servers, has become a promising area of research.

To address the challenges in the context of DT, the distributed learning paradigm of federated learning (FL) offers promising solutions for protecting user privacy and ensuring data security in heterogeneous environments (*Tang & Wang, 2023*; *Sun et al., 2021*; *Rabbani et al., 2024*; *Su et al., 2023*; *Almalki, Alshahrani & Khan, 2024*). FL is a collaborative learning framework that allows participants to train a global model based on their own local datasets. This is achieved by users transmitting updated model parameters to edge servers for aggregation, which are subsequently broadcast to all participants for the next round of local training. To realize high-fidelity DT in FL, a real-time twin pipeline is essential to ensure data synchronization between the physical system and the digital twin model, particularly in latency-sensitive applications such as industrial control, autonomous driving, and remote healthcare. This pipeline leverages efficient edge computing to minimize data transmission latency and incorporates advanced networking technologies, including 5G/6G, time-sensitive networking, and software-defined networking, to optimize communication efficiency and resource allocation. Importantly, although FL primarily involves the transmission of model parameters between aggregation nodes and participants, there remains a potential risk of privacy leakage if attackers are able to infer original data from these transmitted parameters (*Ma et al., 2020*; *Zhang et al., 2024a*). To mitigate such risks, techniques such as secure multi-party computation (*Zhang et al., 2022*), differential privacy (DP) (*Ouadrhiri & Abdelhadi, 2022*), and physical-layer third-party interference (*Wang et al., 2022*) are commonly applied in FL schemes. In *Zhang et al. (2024b)*, the authors proposed a novel edge-based FL architecture, where blockchain is utilized to manage multiple edge nodes for secure collaboration while preserving their privacy. However, the consensus verification process in blockchain introduces significant communication overhead.

The main goal of DP is to prevent inference attacks by introducing noise into sensitive data, thereby masking the results of data queries and protecting individual privacy. While the introduction of significant noise can enhance data privacy, it also significantly reduces the accuracy of the training process for learning models. Additionally, the privacy budget plays a pivotal role in regulating the level of privacy protection, with larger values indicating a more relaxed privacy constraint. Selecting an appropriate privacy budget in heterogeneous environments remains a challenging task (*Ouadrhiri & Abdelhadi, 2022*; *Zhao et al., 2021*). The physical-layer third-party interference scheme employs friendly jammers to transmit artificial noise, aiming to disrupt the eavesdropper's ability to intercept communications. However, this approach reduces the security throughput of edge servers, increases the latency in collecting local updates, and incurs additional costs for interference services, making it less cost-effective (*Wang et al., 2022*; *Xie et al., 2023*). In contrast, homomorphic encryption enables secure multi-party computation by allowing operations to be performed directly on ciphertext. The decryption of the result yields the same value as performing the operation on the plaintext. By using homomorphic encryption in FL, the accuracy of model training remains equivalent to that achieved with plaintext, as the underlying data is never exposed. As a classical homomorphic encryption scheme, Paillier encryption offers superior security and independence in privacy protection compared to DP. DP typically relies on a trusted server to control the noise addition process, which introduces potential risks of noise

manipulation, potentially leading to privacy breaches or model performance degradation. In contrast, Paillier encryption operates independently of the server's trustworthiness, ensuring that even if the server acts as a potential adversary, it cannot directly access the original gradient information of users. This effectively mitigates privacy risks associated with noise manipulation. Furthermore, Paillier encryption enables secure aggregation while preserving computational accuracy, enhancing the security and robustness of FL in distributed environments (*Tang & Wang, 2023*). As such, homomorphic encryption is better suited for FL applications in DT scenarios within heterogeneous environments, where privacy and security are paramount (*Fang & Qian, 2021*; *He et al., 2022*). Furthermore, achieving synchronization in DT systems depends on effectively mapping biases, as the real-world environment is inherently complex. In the context of FL global aggregation, it is crucial to account for mapping frequency and the historical behavior of LDs. This helps to prevent malicious users from submitting erroneous or suboptimal updates to the server, which could compromise the accuracy and reliability of the global model (*Sun et al., 2021*; *Qiao et al., 2024*).

This article explores the application of homomorphic encryption for privacy protection in FL, with a particular focus on its suitability for DT in heterogeneous environments. The analysis evaluates both the accuracy and latency implications of this approach. In our trust-based FL global aggregation scheme, in response to potential DT mapping biases, different end users are assigned varying update weights, with more biased users receiving higher weights. Additionally, learning quality and interaction records are used to track malicious updates, thereby mitigating the threat posed by malicious end users. Since FL global aggregation on the server involves averaging, which can be decomposed into addition and multiplication, Paillier homomorphic encryption is employed to encrypt local training parameters and aggregate the corresponding ciphertexts. However, the encryption stage based on Paillier's algorithm requires two extensive exponential modular multiplication operations, which can be computationally expensive. To address this issue, we introduce a new hyperparameter and precalculate multiple random intermediate values during the key generation phase, simplifying the encryption stage to basic operations and improving efficiency.

The remainder of this article is organized as follows. 'System Model and Preliminaries' presents the system model and and preliminaries. 'Adaptive FL Scheme utilizing an Enhanced Paillier Encryption for Privacy Protection' introduces the adaptive FL scheme utilizing an enhanced Paillier encryption for Privacy Protection. In 'Performance Analysis', we perform a performance analysis of the proposed scheme, focusing on security and privacy, training accuracy, and latency. 'Experimental Evaluation' provides experimental evaluations of the proposed algorithms and scheme, followed by a discussion of the results. Finally, 'Conclusions' summarizes the work presented and outlines potential avenues for future research.

# SYSTEM MODEL AND PRELIMINARIES

This section presents the system model and the preliminary work utilized in this paper.

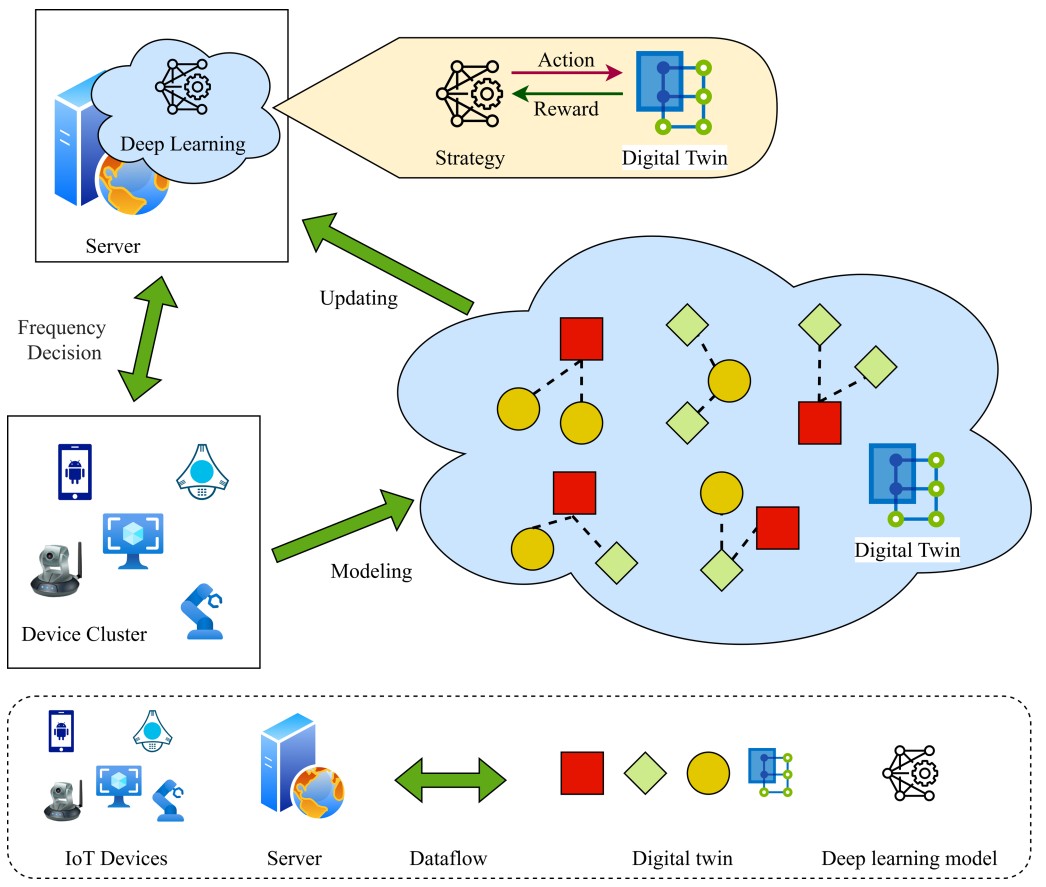

**Figure 1** FL for DT in heterogeneous scenarios.

### DT in FL

As illustrated in Fig. 1, this paper presents the application scenario of FL in DT contexts. This scenario comprises various types of LDs, a management server, and DTs of the LDs. LDs with limited resources communicate with the server *via* wireless communication links. The physical states of the LDs are mapped into virtual digital space to generate DT, which are updated in real-time (*Aloqaily, Ridhawi & Kanhere, 2023*). The DTs of LDs are established by their associated servers, which present the historical and current behaviors of the equipment in digital form by collecting and processing the existing critical physical states of the device. At time $t$, the DT of training node $i$ can be expressed as:

$$DT_i(t) = \{F\left(w_i^t\right), f_i(t), Resource_i(t)\}, \tag{1}$$

where $w_i^t$ represents the training parameter of node $i$ at time $t$, $F\left(w_i^t\right)$ denotes the training state of node $i$ at time $t$, while $f_i(t)$ signifies the CPU frequency of node $i$ at time $t$. Additionally, $Resource_i(t)$ illustrates the resource state of node $i$ at time $t$, encompassing computing power, communication bandwidth, memory capacity, and power capacity.

In light of the possibility of a discrepancy between the mapped values of the DT and the physical device capability characteristics at a specific point in time following mapping, we

examine the deviation between the actual values and the mapped DT values. Accordingly, the calibrated DT model can be expressed as follows:

$$\widetilde{DT}_i(t) = \{F(w_i^t), f_i(t) + \widetilde{f}_i(t), Resource_i(t) + \widetilde{Resource}_i(t)\}, \tag{2}$$

where $\widetilde{f}_i(t)$ is the CPU frequency deviation, which represents the deviation between the actual value of the device and its DT mapping value, and $\widetilde{Resource}_i(t)$ is the resource deviation. It is noteworthy that DT is capable of receiving physical state data from devices and performing self-calibration based on empirical deviation values. This enables the maintenance of consistency with the devices, the provision of authentic feedback information, and the achievement of dynamic optimization of the physical world (*Alzubi et al., 2023*).

## The operation of FL

In practical applications, LDs (such as smartphones, sensors, *etc.*) situated in disparate geographical locations or even under the control of different application service providers must collaborate based on FL to fulfil the requisite application services. As illustrated in Fig. 1, a smartphone equipped with sensors acquires a substantial volume of user data in a real-time monitoring setting. By engaging in collaborative learning and intelligent analysis with clients, the system is able to make more informed decisions regarding quality control and predictive maintenance, obviating the necessity to transmit substantial quantities of real-time user data collected by sensors.

The initial stage of the FL process is the initialization of the task, whereby the manager disseminates the task and the initialised global model, designated $w_0$. Subsequently, upon receipt of $w_0$, the LD $i$ utilizes its data $D_i$ to update the local model parameters $w_{e,i}^t$, thereby identifying the optimal model parameters that minimize the loss function. Accordingly, the local loss function of each LD $i$ can be defined as follows:

$$F(w_{e,i}^t) = \frac{1}{D_i} \sum_{d_i \in D_i} f(w_{e-1,i}^t, d_i), \tag{3}$$

where $t$ denotes the current local iteration index and $e$ represents the current global iteration index. The function $f(w_{e-1,i}^t, d_i)$ quantifies the discrepancy between the estimated value and the true value of the running data instance $D_i$. The variable $d_i$ represents the samples from the training data $D_i$. Upon completion of the $T$-round local training phase, the updated local model parameters are transmitted to the server at a predefined frequency. The server then executes global model aggregation to obtain the parameters for the $e$-th aggregation based on the specified aggregation strategy (for further details, please refer to next section). The loss value after the $e$-th global aggregation is denoted by $F(w_e)$. Subsequently, the server disseminates the revised global model parameters to each node. It is crucial to reiterate the processes of local model training and global model aggregation until the global loss function converges or the model achieves the predetermined level of accuracy.

## Trust-based aggregation in FL

In practical applications, to enhance the learning efficacy of FL and its resilience to malevolent attacks, the parameters uploaded by LDs with high reputation should be

accorded greater weight in global aggregation. In contrast with traditional reputation models, which solely take security threats into account, this model considers the influence of DT bias, learning efficacy, and dataset quality on learning in a comprehensive manner. It is important to note that during the mapping process, deviations in CPU frequency are an inherent consequence of DT. Furthermore, the magnitude of these deviations varies depending on the specific LD in question. It is recommended that LDs with minimal mapping deviations be assigned a greater weight proportion. Moreover, in the context of integrity attacks, malicious nodes may disseminate suboptimal local model updates to the server, thereby compromising the accuracy of the global model. Consequently, incorporating learning quality and interaction records into the calculation of malicious updates can mitigate the impact of malicious clients on the accuracy of the global model. In accordance with the subjective logic model, the belief of server $j$ regarding node $i$ in time slot $t$ can be expressed as follows:

$$B_{i \rightarrow j}^t = \left(1 - \mu_{i \rightarrow j}^t\right) \widetilde{f}_i(t) Q_{i \rightarrow j}^t \frac{a_{i \rightarrow j}^t}{a_{i \rightarrow j}^t + b_{i \rightarrow j}^t}, \tag{4}$$

where $\mu_{i \rightarrow j}^t$ represents the probability of packet transmission failure, $Q_{i \rightarrow j}^t$ represents the learning quality based on the honesty of the majority of LDs, $a_{i \rightarrow j}^t$ is the number of positive interactions, and $b_{i \rightarrow j}^t$ is the number of malicious behaviors (such as uploading false data). In particular, the server employs the FoolsGold scheme (*Yang et al., 2024*), which identifies unreliable clients based on the diversity of local model updates in non-IID FL (*Zhu et al., 2021*), where each node's training data exhibits a distinctive distribution.

Specifically, the FoolsGold scheme is employed to modify the learning rate of each LD in each iteration. This is achieved by distinguishing between honest and dishonest LDs through the use of gradient updates. It is recommended that a client learning rate be maintained which provides unique gradient updates, and that the repetition of client learning rates providing similar gradient updates be reduced. The reputation value of server $j$ corresponding to LD $i$ is represented as follows:

$$R_{i \rightarrow j} = \sum_{t=1}^{T} B_{i \rightarrow j}^t + \tau \mu_{i \rightarrow j}^t, \tag{5}$$

where $i \in [0, 1]$ indicates the degree of uncertainty affecting reputation. In the context of global aggregation, the server is responsible for retrieving updated reputation values and aggregating the local model $w_{e-1,i}^t$ with the participation of LDs into a weighted global model. This can be expressed as

$$w_e = \frac{\sum_{i=1}^{N} \sum_{t=1}^{T} R_{i \rightarrow j}^t w_{e-1,i}^t}{\sum_{i=1}^{N} R_{i \rightarrow j}}, \tag{6}$$

where $w_e$ represents the global parameters resulting from the $e$-th global aggregation, $N$ denotes the number of LDs. By incorporating a trust-based aggregation approach, the inherent bias of DT is accounted for, effectively mitigating the security risks posed by malicious participants while enhancing the framework's resilience and accelerating the learning convergence process (*Wang et al., 2023a*).

### Local model secure upload scheme

In our system model, we utilize the Paillier homomorphic encryption algorithm to ensure the secure uploading of local models by LDs, thereby preventing the leakage of local models. Homomorphic encryption represents a distinctive encryption methodology that enables the processing of ciphertext, with the resulting decryption identical to that of the corresponding plaintext. This guarantees that other users and any third parties who may attempt to gain unauthorized access to the data will be unable to obtain the private information of the data owner (*Yi, Paulet & Bertino, 2014*). Paillier homomorphic encryption is a public-key additive homomorphic encryption scheme comprising the following computational steps (*Fazio et al., 2017*).

(1) Key generation $KeyGen(p,q) \rightarrow (pk,sk)$: Let $p$ and $q$ be two large prime numbers such that their greatest common divisor $gcd(pq,(p-1)(q-1)) = 1$. Then, we can conclude that $n = pq$ and $\lambda = lcm(p-1,q-1)$. Next, a random integer $g \in Z_{n^2}^*$ is selected, and $\gamma = (L(g^\lambda \bmod n^2)) \bmod n$, where $L(x) = (x - \frac{1}{n})$. Therefore, the public key is represented as $pk = (n,g)$, while the private key is represented as $sk = (\lambda,\gamma)$.

(2) Encryption $Enc(\varphi,pk) \rightarrow c$: $\varphi \in Z$ is the plaintext and $r < n$ is a random integer, the ciphertext $c$ can be calculated as $c = g^\varphi \cdot r^n \bmod n^2$.

(3) Decryptio $Dec(c,sk) \rightarrow \varphi$: The plaintext can be decrypted by computing $\varphi = \gamma \cdot L(c^\lambda \bmod n^2) \bmod n$.

## ADAPTIVE FL SCHEME UTILIZING AN ENHANCED PAILLIER ENCRYPTION FOR PRIVACY PROTECTION

In this section, we propose an enhanced Paillier encryption algorithm and introduce our FL-FedDT scheme, which provides privacy protection and latency reduction for FL in DT applications within heterogeneous scenarios.

### Enhanced paillier homomorphic encryption algorithm

The original Paillier algorithm necessitates the execution of a substantial exponential modular multiplication operation on numerous occasions, which inevitably results in the substantial consumption of computational resources at the LD level. Accordingly, we propose an enhanced Paillier encryption algorithm to reduce the computational burden and enhance efficiency while maintaining the integrity of the underlying cryptosystem. The following steps are involved in the computation process:

(1) Key generation $KeyGen(p,q) \rightarrow (pk,sk)$: Let $p$ and $q$ be two large prime numbers such that their greatest common divisor $gcd(pq,(p-1)(q-1)) = 1$. Therefore, we have $n = q$ and $\lambda = lcm(p-1,q-1)$. Next, several values $r \in Z_n^*$ are randomly selected, and compute $f = r^n \bmod n^2$ previously. Additionally, a secret parameter $\xi = \lambda^{-1} \bmod n$ is introduced. Consequently, the public key is defined as $pk = n$, while the secret key is defined as $sk = (\lambda,\xi)$.

(2) Encryption $Enc(\varphi,pk) \rightarrow c$: $\varphi \in Z_n$ is the plaintext and a randomly selected value of $f$ is used to compute the ciphertext, which is given by $c = (1 + \varphi n) \cdot f$.

(3) Decryption $Dec(c,sk) \rightarrow \varphi$: The plaintext can be decrypted by computing $\varphi = \xi \cdot L(c^\lambda \bmod n^2) \bmod n$.

In the enhanced Paillier encryption algorithm, the parameter $g$ for public key $pk$ is disregarded, and a confidential parameter $\xi$ is incorporated to substitute for $\gamma$ in the initial phase of key generation. This procedure is advantageous for the encryption phase. As a consequence of the fact that multiple have already been computed, the encryption process requires only basic mathematical operations, rather than the more complex exponential and modular multiplication operations. This results in a significant reduction in the time required for the calculation.

As demonstrated in the subsequent analysis, the enhanced Paillier encryption algorithm continues to exhibit the homomorphic supplementary attribute characteristic of homomorphic encryption. For plaintext $\alpha$ and $\beta$, we have

$$Enc(\alpha) = (1 + n\alpha) \cdot r_1^n \setminus mod\ n^2, \tag{7}$$

$$Enc(\beta) = (1 + n\beta) \cdot r_2^n \setminus mod\ n^2. \tag{8}$$

Subsequently, we can obtain

$$\begin{aligned} Enc(\alpha) &\bigotimes Enc(\beta)\, mod\ n^2 \\ = \left[(1+n\alpha) \cdot r_1^n\right] &\bigotimes \left[(1+n\beta) \cdot r_2^n\right] mod\ n^2 \\ = \left[1 + n(\alpha+\beta)\right] &\cdot (r_1 r_2)^n\, mod\ n^2 \\ &= Enc(\alpha+\beta)\, mod\ n^2. \end{aligned} \tag{9}$$

Therefore, $E(\alpha) \bigotimes E(\beta) = E(\alpha+\beta)$ holds. Similarly, the decryption process can be obtained using the same method, expressed as $Dec\left(Enc(\alpha) \bigotimes Enc(\beta)\right) = \alpha + \beta$.

### Privacy protection algorithm for FL

The operational workflow of the proposed FL framework and the details of homomorphic encryption applied during local training are presented in Figs. 2 and 3. In particular, upon completion of the local training phase, the LDs will then proceed to homomorphically encrypt the uploaded local model. Subsequently, the server will collate all encrypted weights and perform global aggregation in accordance with Eq. (6). The aggregated encrypted weights can be represented as follows:

$$[[w_e]] = \left[\left[\frac{\sum_{i=1}^{N}\sum_{t=1}^{T}R_{i \to j}^{t}w_{e-1,i}^{t}}{\sum_{i=1}^{N}R_{i \to j}}\right]\right] \tag{10}$$

where $[[\cdot]]$ represents the data element designated homomorphic encryption and $N$ denotes the number of LDs. Considering the additional characteristics of homomorphic encryption, the global weight of the encryption can be computed as follows:

$$[[w_e]] = \left[\left[\frac{\sum_{i=1}^{N}\sum_{t=1}^{T}R_{i \to j}^{t}w_{e-1,i}^{t}}{\sum_{i=1}^{N}R_{i \to j}}\right]\right] = \frac{\sum_{i=1}^{N}\sum_{t=1}^{T}R_{i \to j}^{t}[[w_{e-1,i}^{t}]]}{\sum_{i=1}^{N}R_{i \to j}}. \tag{11}$$

It is important to note that in order to ensure the integrity and security of data communication, the use of the Transport Layer Security/Secure Sockets Layer protocol is essential in the communication channel between the server and LDs.

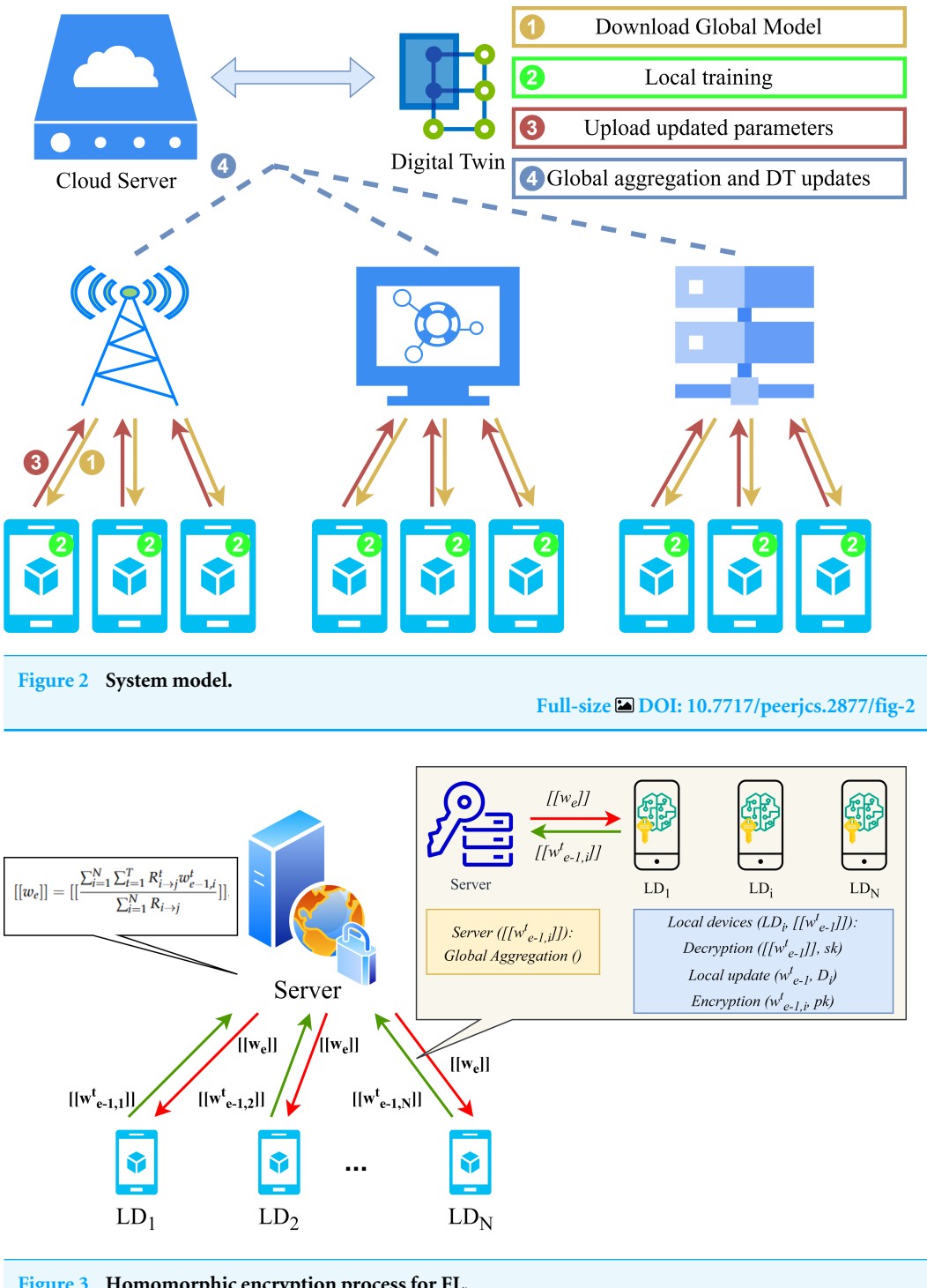

**Figure 2   System model.**

**Figure 3   Homomorphic encryption process for FL.**

In our enhanced Paillier homomorphic encryption algorithm, we have completed the fundamental operations of selecting a random integer *r* for large exponentiation and modular multiplication in each iteration of the original Paillier algorithm in advance,

during the key generation stage. Consequently, the complete encryption process remains unaltered, and both algorithms possess an equivalent level of security and semantic security, as well as the capacity to withstand plaintext attacks. This renders our solution resilient to attacks perpetrated by malevolent actors. Furthermore, the enhanced Paillier algorithm prioritizes reducing computational latency. However, if key pairs are generated with each iteration, the latency will be significantly increased. Accordingly, a suitable key length is selected for the purposes of ensuring security and low latency in the context of experimentation.

## A privacy protection and low latency FL scheme for DT

Figure 2 illustrates the collaborative interaction process of our privacy protection algorithm between the server and LDs. Algorithms 1 and 2 offer a synopsis of the adaptive FL scheme that we have devised and which is based on the enhanced Paillier encryption.

The server's primary function is to aggregate the updated encryption weight $[[w_e]]$ on a global scale and to execute the $e$-th FL iteration process. There are $N$ LDs that participate in this process. Subsequently, the LD will decrypt the aggregated weight $[[w_e]]$, perform local training, and encrypt the updated weight for the subsequent round of $e + 1$ FL iteration.

### Global aggregation

In the heterogeneous scenario of FL used for DT, we assume that all edge users are honest but curious. This implies that they may be curious about the information stored on LDs and may attempt to infer privacy information through parameters. Concurrently, third-party malevolent eavesdroppers may also obtain user privacy through reverse engineering gradient attacks and other methods. Each edge server is equipped with information on all user lists within its communication range.

In each round $e$ of FL iteration, an adaptive FL scheme is employed in collaboration with the privacy protection algorithm. In the initial phase of the learning process at FL iteration round $e = 0$, the server first initializes a learning model $F(w_e)$, encrypts the initial global weight $w_0$ with a public key, and transmits it to all $N$ LDs. The initialisation of the local models and reputation parameters that are to be trained for the LDs is required. Moreover, each LD decrypts the global weight utilized for local training and returns the updated weight $w_{e-1,i}^t$ after $T$ rounds. Subsequently, the LD transmits the encrypted weight $[[w_{e-1,i}^t]]$, together with the public key, to the server. In the case where $e > 0$, the server updates the aggregated global weight $[[w_e]]$ through the application of Eq. (10) following the receipt of all weight updates from the preceding FL iteration process. It should be noted that this step reflects the additive homomorphic property of homomorphic encryption algorithms as described in Eq. (9). In the course of successive rounds $e$ and $e + 1$, the count time slot and the time threshold are designated as $\varepsilon$ and $\Delta$, respectively. Upon receipt of the parameters by the server and $\varepsilon < \Delta$, the FL process resumes and advances to the subsequent round. In the event that the aforementioned conditions are not met, the algorithm will exit the current iteration and set the current encrypted global weight to the final value. The final weight will be the result of the last global aggregation, occurring after $E (E > 1)$ global rounds without interruption. Ultimately, the final aggregated model must

be transmitted to all $N$ LDs, where it will be used to update the local model and conduct an actual evaluation.

### Local model updating

Upon receipt of the $[[w_e]]$ from the server, the LDs undertake the decryption operation of the enhanced Paillier algorithm. By employing the private key $sk$ of each LD, the value $w_e$ can be obtained and assigned to $w_{e-1,i}^t$. Subsequently, the model is updated using the small batch gradient descent algorithm, encrypted with the public key $pk$ to obtain $[[w_{e,i}^t]]$, and uploaded to the server.

## PERFORMANCE ANALYSIS

In application scenarios such as healthcare and military systems, security is typically prioritized over other performance metrics, such as latency in heterogeneous environments. These scenarios involve highly sensitive data, such as patient privacy and classified military intelligence, making data security more critical than minimizing communication and computation latency. However, in other domains like industrial IoT, the trade-off between latency and security becomes particularly significant. For instance, in smart manufacturing and remote medical monitoring, low latency is essential for real-time decision-making, whereas stringent security measures may introduce additional computational and communication overhead, potentially increasing system response time. Therefore, achieving an optimal balance between security and real-time performance is crucial, depending on the specific application requirements (*Tang et al., 2024*; *Tang et al., 2025*). Although our proposed solution may incur higher communication and computational costs, it demonstrates superior privacy protection compared to baseline methods while ensuring that the final model maintains optimal training accuracy, thereby achieving a well-balanced trade-off between data security and model performance.

### Security and privacy performance analysis

In our proposed privacy protection scheme, we utilize the Paillier homomorphic encryption algorithm to safeguard privacy information, circumvent plaintext selection attacks, and guarantee semantic security. Specifically, the parameter communication and aggregation are trained in ciphertext form, ensuring that no information from the plaintext is leaked. The security of Paillier encryption is contingent upon the computational complexity of integer factorization problems, particularly the Decisive Composite Residual Assumption on the composite residual group. Even if an adversary gains access to the public key and random parameters, it is challenging to derive any information about the plaintext from the ciphertext, thereby ensuring the confidentiality of the data. To date, no polynomial-time algorithm has been able to successfully break the encryption. Furthermore, Paillier encryption incorporates random numbers into each encryption operation, ensuring that identical plaintext will yield disparate ciphertext outputs. The introduction of random numbers into the Paillier encryption process ensures semantic security, which means that it is difficult to distinguish between different plaintexts when the ciphertext is known. The incorporation of this random number also serves to effectively prevent replay attacks,

---

**Algorithm 1** The Global Aggregation of Our Proposed FL Scheme

---

**Input:** The weight $\left[\left[w_{e-1,i}^{t}\right]\right]$ of the encryption for each LD;

Local training rounds $T$;
Time threshold $\Delta$;
Global aggregation rounds $E$.

**Output:** Global weight $[[w_e]]$ of encryption.

1: The global model $F(w_e)$ should is initialized with a global weight of $w_0$.

2: Initialize all $N$ LDs and set the reputation parameters.

3: **while** $e < E$ **do**

4: **if** $e = 0$ **then**

5: The global weight $w_0$ is to be encrypted using the public key in our enhanced Pailler encryption algorithm, resulting in $[[w_0]]$.

   6: $e \leftarrow 1$.

7: **else**

  8: **for** each LD **do**

    9: In accordance with Algorithm 3, the server receipt of training update weight $\left[\left[w_{e-1,}^{t}\right]\right]$ from each LD.

  10: **end for**

11: **end if**

12: $[[w_e]] \leftarrow \dfrac{\sum_{i=1}^{N}\sum_{t=1}^{T}R_{i\rightarrow j}^{t}\left[\left[w_{e-1,i}^{t}\right]\right]}{\sum_{i=1}^{N}R_{i\rightarrow j}}$. //According to (10), utilizing the additive property of homomorphic encryption for global aggregation.

13: Send $[[w_e]]$ to all $N$ LDs.

14: Update all reputation parameters according to (5).

15: Counting time slot $\varepsilon$.

16: **if** $\varepsilon > \Delta$ **then**

  17: *Mark* = *Stop*.//Interrupt signal.

  18: Send *Mark* to all N LDs.

  19: Break.

20: **end if**

21: **end while**

22: Send $[[w_e]]$ to all $N$ LDs.

---

thereby ensuring that attackers are unable to analyze the data content by repeatedly observing the ciphertext. The proposed enhanced Paillier encryption algorithm represents an improvement on the original algorithm, offering the same level of security and sufficient protection against attacks from malicious actors.

For servers that are both honest and curious, the updated parameters that are received from LDs are in ciphertext form. Furthermore, the global aggregation algorithm is an addition operation, which means that all operations are homomorphic and there is no

---

**Algorithm 2** The Local Training of Our Proposed FL Scheme

---

1: **for** all $N$ LDs **do**
2:  Receive *Mark* and encrypted global weight $[[w_e]]$ from the server.
3:  **if** $Mark = Stop$ **then**
4:   Break.
5:  **else**
6:   The LDs utilize their private key *sk* to decrypt the global weight and obtain $w_e$.

7:   $w_e$ is assigned to $w_{e-1,i}^t$ and using the foolsGold scheme to get $w_{e,i}^t$ after $T$ local training rounds.

8:   Each LDs utilize its public key *pk* to encrypt $w_{e,i}^t$.
9:   Upload $\left[\left[w_{e,i}^t\right]\right]$ to the server *via* communication protocol.

10:   **end if**
11:  **end for**

---

decryption process in the server at this stage. Consequently, all data stored on the server is encrypted. Consequently, even in the event of interaction between the server and third-party entities, the absence of a private key renders the acquisition of any available information impossible. Furthermore, the LD possesses its own private key, which it utilizes in conjunction with the public key to execute the enhanced Paillier encryption algorithm. This results in the transmission of encrypted local updates to the server. Following each round of global aggregation on the server, the updated global weights are obtained. However, the result remains in an encrypted format. Subsequently, the data must be decrypted using the private key of the LD. To enhance the security of the system, it is possible to set a larger key length or generate key pairs for each round. Throughout the entirety of the learning process, LDs are unable to access data from other devices.

Furthermore, our global aggregation is founded upon the principle of trust. Parameters uploaded by LDs with a high reputation are accorded greater weight in the global aggregation, thereby preventing malicious LDs from providing erroneous or inferior updates to the server in the event of a Byzantine attack. This ultimately serves to diminish the precision of the global model.

### Training accuracy performance analysis

In our proposed solution, we utilize an enhanced Paillier algorithm to achieve global aggregation based on homomorphic encryption, with the objective of enhancing privacy and security. In the FL process based on homomorphic encryption, the data is only encrypted without any distortion operation, thereby ensuring that the training accuracy remains almost unchanged. As indicated in Section 'Experimental Evaluation', the results of the performance evaluation demonstrate that the accuracy remains at a high level.

## Latency performance analysis

In our FL system employing trust-based global aggregation, the latency in the $e$-th round consists of three primary components: $T_e^{\text{Server}}$, the computation time for aggregating global weights on the server; $T_e^{\text{LD}}$, the time required for local training on each device using its dataset; and $T_e^{\text{Com}} = T_e^{\text{up}} + T_e^{\text{down}}$, the communication time between the server and LDs, where $T_e^{\text{up}}$ and $T_e^{\text{down}}$ represent the upload and download times, respectively.

For the purposes of more rigorous analysis, it is assumed that the $T_e^{\text{LD}}$ and $T_e^{\text{Com}}$ times of each LD are identical. In practice, data that is unevenly distributed across LDs will result in differing computation times $T_e^{\text{LD}}$ and communication times $T_e^{\text{Com}}$ depending on the distance between the server and the LD. In consequence, the overall waiting time for FL in each round $e$ is given by

$$T_e^{\text{FL}} = T_e^{\text{Server}} + T_e^{\text{LD}} + T_e^{\text{Com}}. \tag{12}$$

Nevertheless, the use of enhanced Paillier homomorphic encryption in FL serves to enhance privacy and security in heterogeneous environments. It is evident that the additional encryption and decryption operations will result in an increase in computation time. In particular, LDs encrypt local training parameters and transmit them to the server. They also decrypt global weights for the subsequent training round after receiving aggregated weight from the server. In consequence, the time required for local training on LDs can be expressed as follows:

$$T_e^{LD,Pai} = T_e^{LD} + T_e^{Enc} + T_e^{Dec}, \tag{13}$$

where $T_e^{\text{Enc}}$ denotes the encryption computation time using a public key, and $T_e^{\text{Dec}}$ denotes the decryption computation time with a private key. Since key generation occurs only once at $e = 0$, the computation time for key pair generation is excluded from the scheme's overall time analysis.

Furthermore, $T_e^{\text{Server}}$ and $T_e^{\text{LD}}$ can be regarded as analogous to the baseline scheme based on trusted global aggregation FL, given that the data utilized for computation and transmission is ciphertext rather than plaintext. In consequence, the overall latency of our FL system with homomorphic encryption at each round $e$ can be expressed as follows:

$$T_e^{FL,pai} = T_e^{Server,pai} + T_e^{LD,pai} + T_e^{Com,pai} = T_e^{Server} + T_e^{LD} + T_e^{Com} + T_e^{Enc} + T_e^{Dec}. \tag{14}$$

It is evident that the encryption and decryption processes, which require a longer latency, result in a greater latency when using encryption schemes compared to non-encryption schemes, *i.e.*, $T_e^{\text{FL}} < T_e^{\text{FL,Pai}}$.

In order to guarantee the security of the system, the enhanced Paillier encryption algorithm has been applied to FL-FedDT. This ensures that the updated weights are not tampered with during communication and reduces the time consumed by encryption, thereby improving the efficiency of the FL system. The original Paillier encryption and the enhanced Paillier encryption are, in essence, homomorphic encryption. Accordingly, the latency performance difference is designated as between the two algorithms is $T_e^{\text{LD,Pai}}$ in Eq. (14), and we can represent their latency performance as $T_e^{\text{LD,Pai,Orig}}$ and $T_e^{\text{LD,Pai,Enh}}$, respectively. The original Paillier encryption process is characterized by

**Table 1  Simulator settings and parameter configuration.**

| Simulation parameter | Numerical value |
| --- | --- |
| Number of LDs $N$ | 100 |
| Local training rounds $T$ | 5 |
| Training model | CNN |
| Data set | MINIST |
| Gradient update method | FoolsGold |
| Size of simulation area | 1,000 m × 1,000 m |

extensive exponential and modular multiplication operations, which contribute to a notable latency. In contrast, the enhanced Paillier encryption process necessitates only fundamental operations, thereby reducing the associated latency. In regard to the decryption process, the two algorithms engage in comparable operations, and thus, a comparison of their decryption computation latency can be ignored. The results of the latency performance evaluation in the fifth section corroborate our hypothesis, i,e., $T_e^{\text{LD,Pai,Orig}} > T_e^{\text{LD,Pai,Enh}}$.

## EXPERIMENTAL EVALUATION

In this section, we conducted a series of comprehensive simulation experiments to assess the efficacy of our proposed FL-FedDT scheme and homomorphic encryption process in heterogeneous environments. Table 1 provides a summary of the parameter configuration and simulation settings. All the results are obtained with a PC of Intel(R) Xeon(R) CPU E5-2670v2 @2.50 GHz.

In order to safeguard the confidentiality of data from LDs utilized for DT mapping in heterogeneous environments, we have incorporated homomorphic encryption into FL. Nevertheless, this will result in an increased computational burden and a concomitant increase in system latency. Accordingly, to reduce the overall processing time, we propose an enhanced Paillier encryption scheme. The Paillier process is implemented using the PyCryptodome library in Python, as this library provides high-level large integer operations suitable for the implementation of custom Paillier algorithms.

### Time for encryption and decryption

In this subsection, encryption tests are conducted using a sample with varying key lengths. As shown in Fig. 4, the encryption time of the original Paillier algorithm is longer than that of the enhanced Paillier algorithm. Moreover, the time difference between the two algorithms increases as the key length increases from 128 to 2,048 bits. This can be attributed to the fact that the original Paillier algorithm requires two extensive exponential modular multiplication operations and one modular multiplication operation for encryption, while the enhanced Paillier algorithm only involves two basic operations. The time difference between the two algorithms is approximately 1.432 ms for a 768-bit key and 4.416 ms for a 1,024-bit key. In terms of decryption, both algorithms exhibit similar performance. Notably, the enhanced Paillier algorithm requires more time for key generation compared to the original, due to the use of multiple random numbers $r$ in the key generation process.

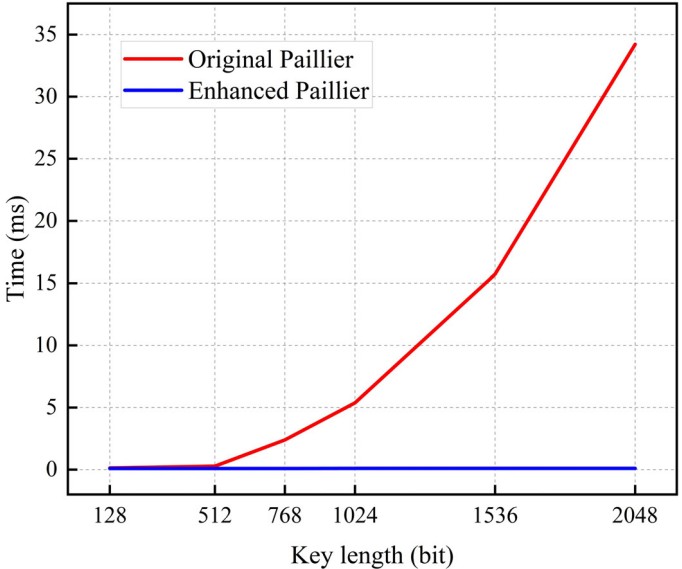

A

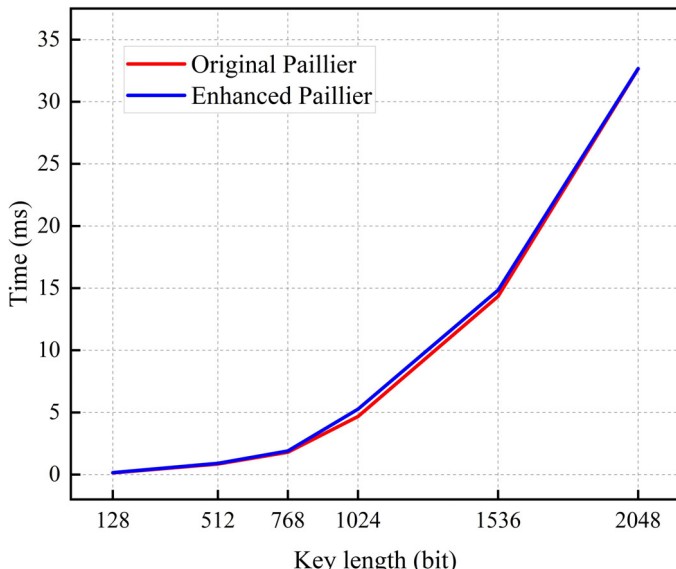

B

**Figure 4 Time for encryption and decryption.** (A) Time for encryption. (B) Time for decryption.

However, the time required for key pair generation is measured in milliseconds. The results indicate that the enhanced Paillier algorithm offers better performance in handling large datasets compared to the original Paillier algorithm.

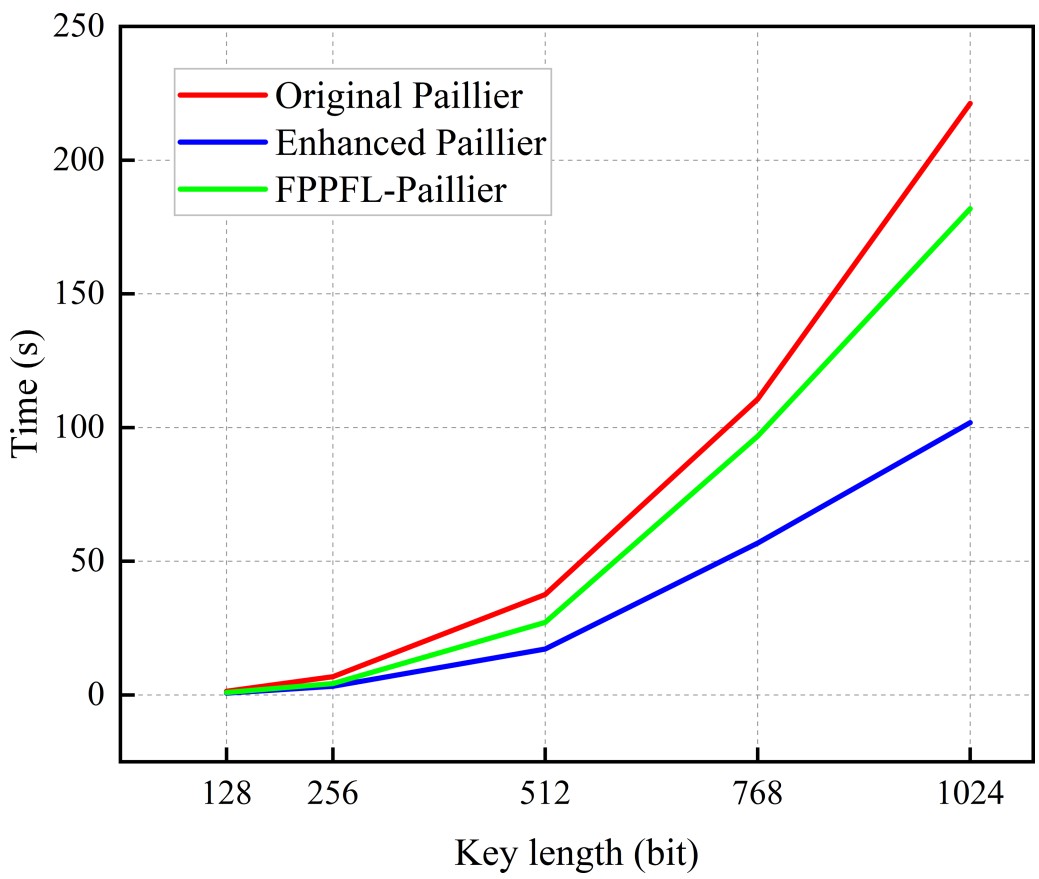

**Figure 5** Time for conducting a round of local training.

## Time for conducting A round of local training

In order to assess the efficacy of the algorithm in processing multiple samples, we extracted the time required for a single iteration of LD training, with the objective of displaying the results of the convolutional neural network (CNN) model and the MNIST dataset (*Alzubi et al., 2023*). As illustrated in Fig. 5, for key lengths within the range of 128 to 1024 bits, the enhanced Paillier algorithm exhibits a reduced execution time in comparison to the original Paillier algorithm. In particular, the time difference for a 512-bit key length is 20.387 s, while for a 1,024-bit key length, it is 119.387 s. Figure 4 illustrates that the time difference increases with the increase of key length in a single round. It is important to note that in practice, there are $N$ LDs performing $E$ rounds of global aggregation, with $T$ rounds of local training conducted on each device. The time cost for each LD in each round is approximately consistent, which will result in a significant delay in achieving optimal security performance. At the same time, compared with the FPPFL-Paillier method in *Tang & Wang (2023)*, we achieved shorter key generation time by simplifying the modulo operation. It can thus be concluded that the enhanced Paillier encryption algorithm has the potential to reduce latency and improve system efficiency.

**Table 2  Key randomness test results.**

| Test items | Original paillier | Enhanced paillier |
|---|---|---|
| Approximate entropy | 1.00, Pass | 1.00, Pass |
| Serial | 0.518631, Pass | 0.511002, Pass |
| Universal | 0.367181, Pass | 0.369127, Pass |
| Runs | 0.264885, Pass | 0.257981, Pass |
| Cumulative sums | 0.225366, Pass | 0.234321, Pass |
| Blook frequency | 0.212557, Pass | 0.202374, Pass |
| Frequency | 0.337311, Pass | 0.329324, Pass |
| Linear complexity | 0.178035, Pass | 0.167439, Pass |
| Longest run | 0.297934, Pass | 0.287161, Pass |
| Non-overlapping template | 0.198821, Pass | 0.184822, Pass |
| Random excursions | 0.209381, Pass | 0.201319, Pass |
| Random excursions variant | 0.034465, Pass | 0.033157, Pass |

## Key randomness analysis

The field of randomness testing pertains to the examination of the randomness of sequences generated by a random number generator or encryption algorithm, employing probability and statistical methodologies. The "Special Publication 800-22" test package, provided by the National Institute of Standards and Technology (NIST) in the United States, is referred to as NIST randomness testing (*Pareschi, Rovatti & Setti, 2007*). These tests can verify the randomness of any long binary sequence generated by hardware and software, and can be employed as a secret random number generator or a pseudo-random number generator. The principal objective of the testing process is to ascertain the presence of any non-random elements within the sequence. The test suite comprises a number of tests, each of which returns a $p$-value. When $p$ is within the range of 0.01 to 1, the binary sequence is deemed to have passed the corresponding test. A higher $p$-value indicates a greater degree of randomness in the sequence. Given that both encryption algorithms generate private keys in a similar manner, it is only necessary to conduct randomness tests on the generated public keys. To ascertain the average value, 100 tests were conducted to simulate the generation of public keys of varying lengths. The results are presented in Table 2. The results demonstrate that the randomness of the binary sequences generated by the two algorithms is essentially identical, thereby indicating that these two algorithms possess an equivalent level of security.

## Accuracy and loss

In our FL-FedDT scheme, a homomorphic encryption algorithm is employed to process model parameters, ensuring the security of the data communication process. Leveraging the additive property of homomorphic encryption, the plaintext remains intact after decryption, thereby preserving the accuracy of the model. As shown in Fig. 5, the accuracy of FL-FedDT-Paillier and FL-FedDT-Paillier-Enhanced is nearly identical to that of the trust-based global aggregation FL scheme. In Fig. 6, FL-FedDT-Paillier represents the trust-based global aggregation FL scheme combined with the original Paillier algorithm,

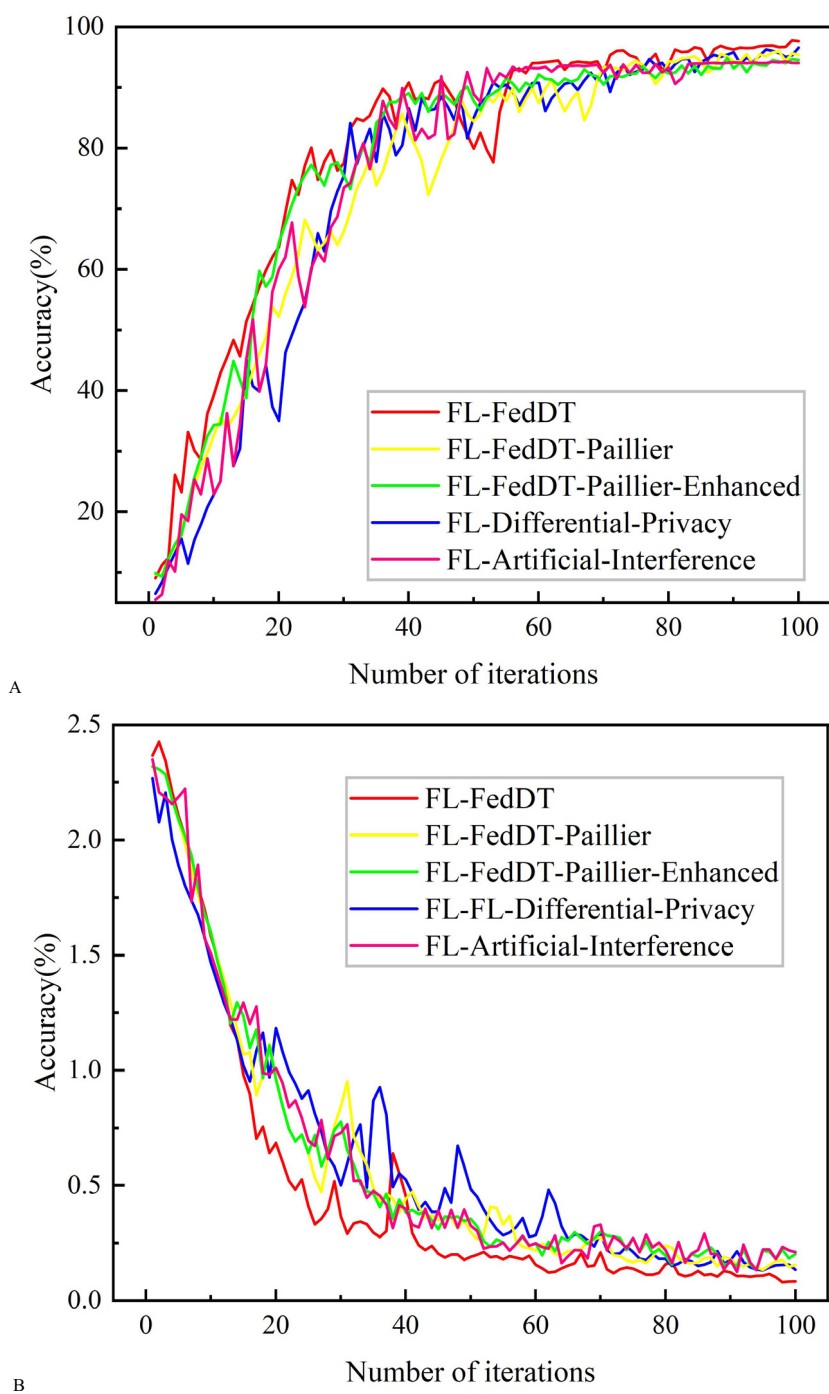

**Figure 6 Testing accuracy and loss comparison with non-IID data.** (A) Testing accuracy. (B) Testing loss.

while FL-FedDT-Paillier-Enhanced represents the trust-based global aggregation FL scheme combined with the improved Paillier algorithm proposed in this work. For ease of calculation, the key length is set to 256 bits, and the global aggregation frequency is

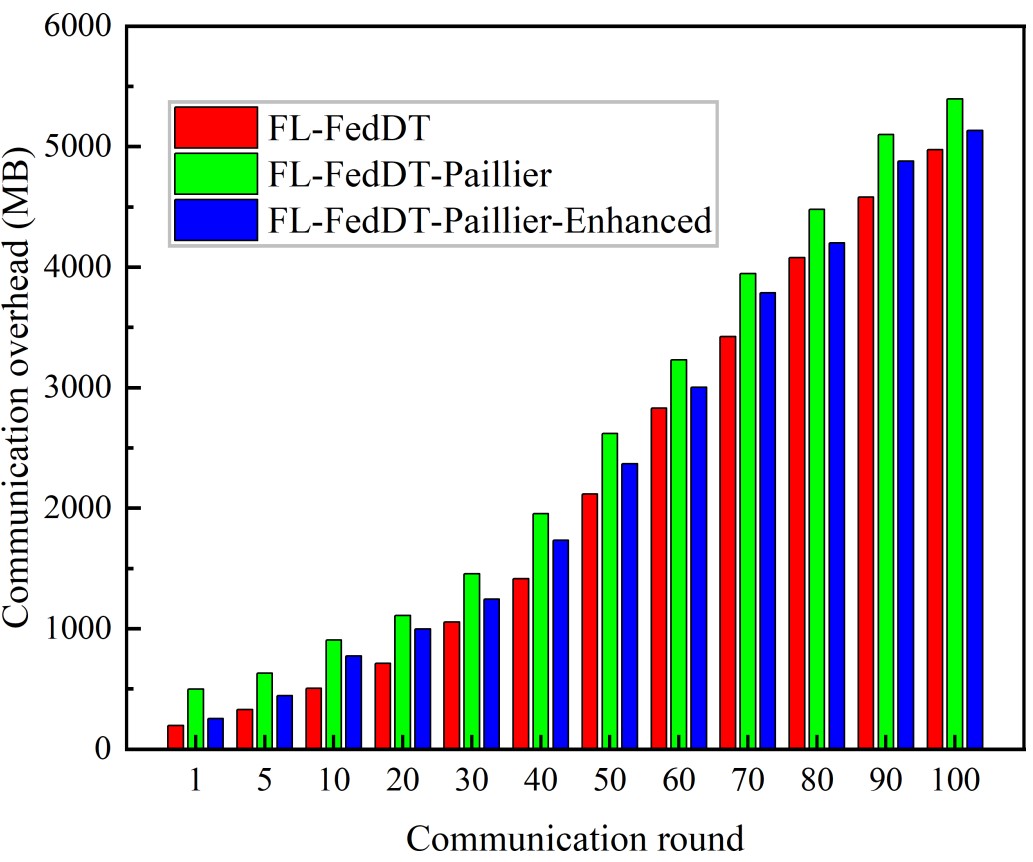

**Figure 7** **The communication overhead accumulated by FL in different communication rounds.**

set to $E = 100$ iterations. Using the default settings in Table 1 and training with non-IID data (*Zhu et al., 2021*), the model achieves an accuracy of approximately 95.38%, with a loss rate reduced to 0.156. Additionally, to highlight the advantages of our approach, we compare it with FL schemes based on differential privacy (*Hu et al., 2020*) and artificial collaborative interference (*Wang et al., 2022*). The differential privacy method protects privacy by adding noise to locally updated weights, with the privacy level determined by the privacy budget. However, it requires more time to converge to the desired accuracy. On the other hand, artificial collaborative interference enhances the security throughput of local devices by enabling cooperation among devices to send jamming signals that disrupt eavesdroppers' wireless links. Nevertheless, this method consumes energy resources intended for local training and communication, thereby reducing training accuracy and increasing communication time. In Fig. 7, we present the communication overhead of parameter transmission between local devices and servers, which shows that our proposed enhanced Paillier encryption scheme has advantages over traditional Paillier encryption schemes.

In addition, Fig. 8 shows the comparative results of training with IID data. The test results show that the training accuracy of the four systems converges, with an accuracy rate close to 98.723%, which is slightly higher than the training with non-IID data, and the

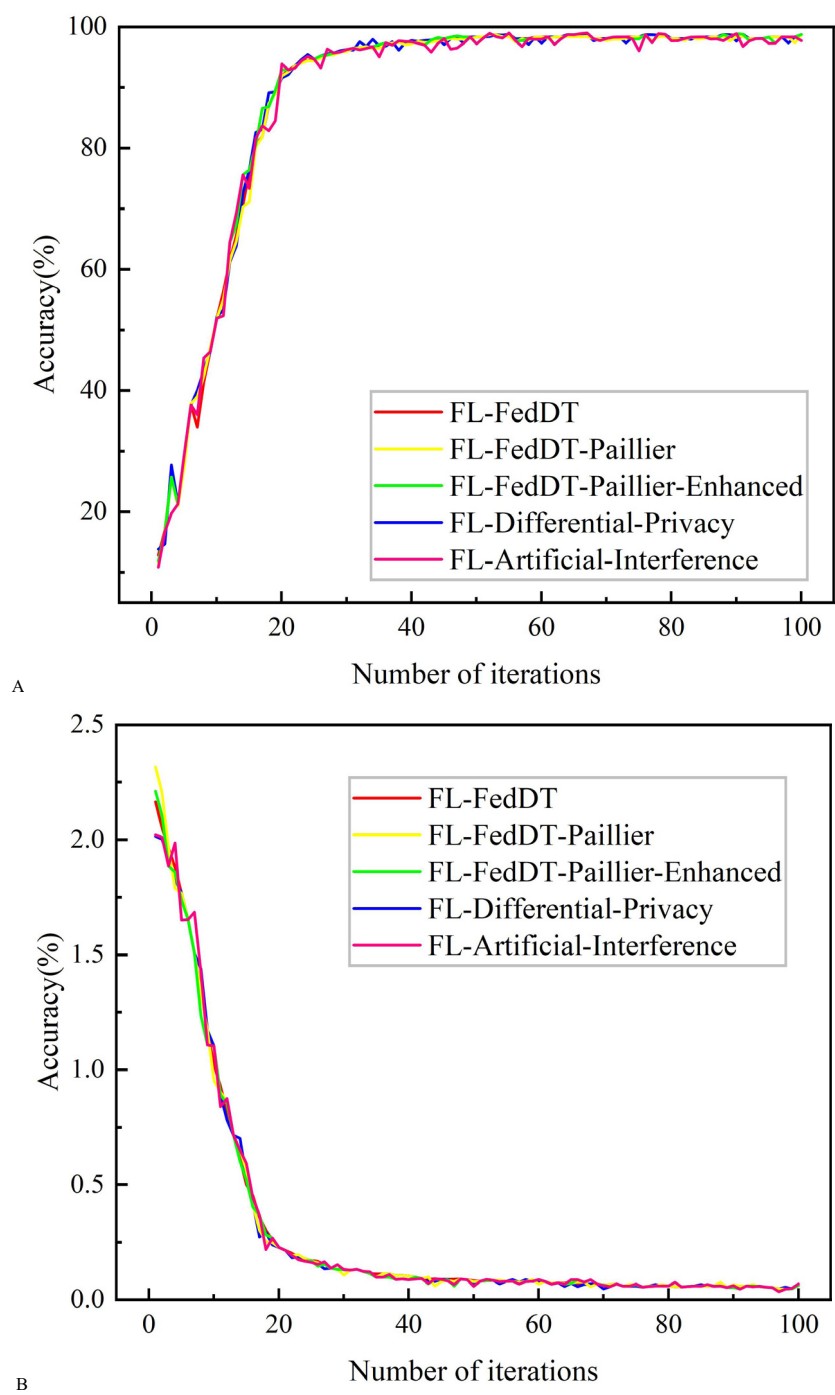

**Figure 8** **Testing accuracy and loss comparison with IID data.** (A) Testing accuracy. (B) Testing loss.

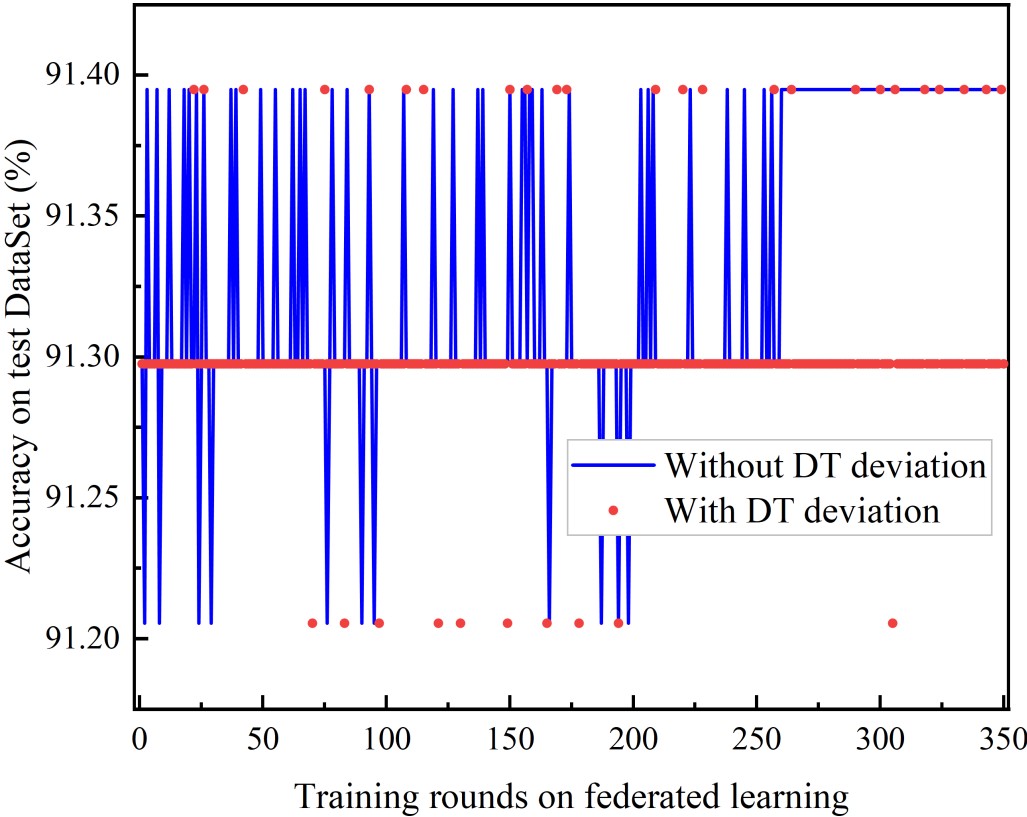

**Figure 9** Comparison of DT deviation and without DT.

curve trends are similar. This is because the IID data are independently extracted from the same distribution for each local device data, so the data from different local devices have the same statistical characteristics. Table 3 presents the convergence time for achieving accuracy in the three FL approaches.

The comparison of joint learning accuracy under the presence of DT bias and after DT bias calibration is shown in Fig. 9. The DT bias calibration FL based on the trust-weighted aggregation strategy achieves better learning performance than the FL with DT bias calibration. Even before convergence, federated learning with bias calibration demonstrates superior performance. Moreover, it is observed that the Deep Q Network (DQN) with DT bias does not converge.

Figure 10 compares the accuracy of joint learning based on DQN with that of fixed-frequency joint learning. As DQN gradually converges, the accuracy of DQN-based federated learning exceeds that of the fixed-frequency federated learning. This is because the global aggregation gain for federated learning accuracy is nonlinear, and the fixed-frequency scheme can avoid aggregation in certain rounds, resulting in lower energy consumption and better learning performance.

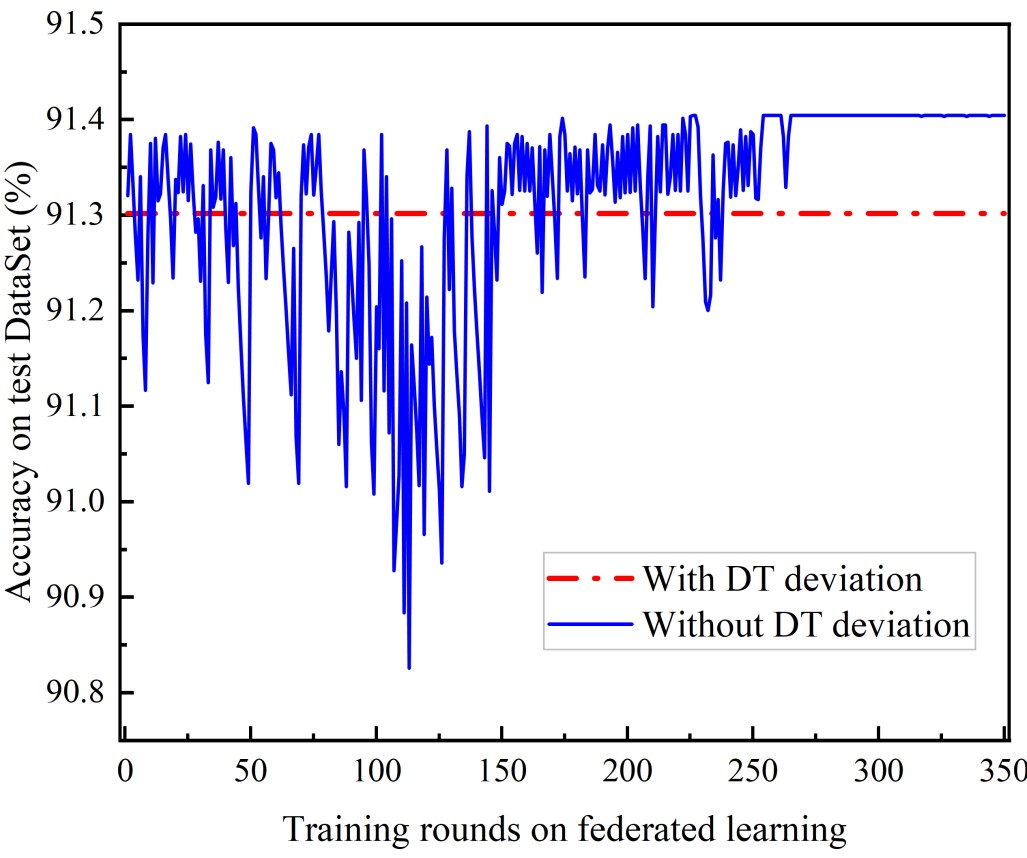

**Figure 10** Comparison of the accuracy achieved of federated learning between adaptive frequency and fixed frequency.

## CONCLUSIONS

This article proposes FL-FedDT, a privacy-preserving and low-latency FL scheme for DT applications, employing enhanced Paillier encryption in heterogeneous environments. To address implicit bias and the risk of malicious nodes uploading low-quality updates during DT synchronization, we propose a trusted global aggregation FL framework. This approach adjusts the aggregation weights of uploaded parameters based on DT mapping bias and interaction records from local devices. Additionally, Paillier homomorphic encryption is employed to safeguard local training parameters against data leakage. Given the extended encryption times and the need for $E$ global iterations inherent in the original Paillier encryption algorithm, this paper introduces an enhancement: a novel hyperparameter $\xi$ and the precalculation of $f = r^n mod n^2$ during the key generation phase. This reduces the complex exponential modular multiplication to simpler operations at the encryption stage. By generating a single key pair and encrypting parameters only once, the time required for model training is significantly decreased. Our analysis and experimental results indicate that the enhanced Paillier encryption algorithm in this scheme provides robust privacy protection. Furthermore, it maintains FL training accuracy comparable to

**Table 3 Time comparison for training one round with varying epochs on LDs.**

| Time (s) | $T = 1$ | $T = 2$ | $T = 3$ |
|---|---|---|---|
| FL-FedDT | 29.27 | 41.97 | 85.66 |
| FL-FedDT-Paillier | 147.13 | 158.39 | 198.62 |
| FL-FedDT-Paillier-Enhanced | 99.41 | 114.39 | 206.13 |

that of the original algorithm ensemble while reducing latency. Nevertheless, there remain several challenges that need to be addressed in the context of larger-scale networks and heterogeneous environments. Resource-constrained edge devices may be unable to support the computational overhead of encryption tasks, such as temperature sensors in healthcare scenarios or operational tracks in industrial settings, which are typically limited to basic data collection and communication functions. As a result, it becomes essential to consider secure communication mechanisms at the physical layer to provide data sources for FL. In the future, exploring collaborative interference models between different devices could optimize FL efficiency while ensuring secure data transmission.

## Funding
The authors received no funding for this work.

## Competing Interests
The authors declare there are no competing interests.

## Author Contributions
- Jie Li conceived and designed the experiments, performed the experiments, analyzed the data, performed the computation work, prepared figures and/or tables, authored or reviewed drafts of the article, and approved the final draft.
- Dong Wang conceived and designed the experiments, performed the experiments, analyzed the data, performed the computation work, prepared figures and/or tables, authored or reviewed drafts of the article, and approved the final draft.

## Data Availability
The code is available at GitHub: https://github.com/fujianU/.

The dataset is available at GitCode: https://gitcode.com/Resource-Bundle-Collection/d47b0/overview?utm_source=pan_gitcode&index=top&type=href&;.

The code is available at Zenodo: fujianU. (2024). fujianU/federatinglearningPaillier: papercode. Zenodo. https://doi.org/10.5281/zenodo.15551053.

## Supplemental Information
Supplemental information for this article can be found online at http://dx.doi.org/10.7717/peerj-cs.2877#supplemental-information.

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
