# Peer review of "Federated learning for digital twin applications: a privacy-preserving and low-latency approach"

_PeerJ Computer Science, doi:10.7717/peerj-cs.2877_

## Round 0.1 · original submission · Major Revisions

The reviewers see merit in the paper, but also identify critical problems to address. The authors are invited to revise their paper according to reviewers' comments.

·

Basic reporting

Language and Clarity: The manuscript uses clear, professional, and unambiguous English throughout. Technical terms are appropriately defined, and the narrative is well-structured.
Literature References: The article provides a comprehensive background, citing relevant and recent literature (e.g., on federated learning, homomorphic encryption, and digital twins). However, the authors should ensure uniform formatting in the reference section for consistency.
Figures and Tables: Figures and tables are relevant, high quality, and well-labeled. They effectively support the claims made in the manuscript. Raw data files were shared, and the associated code was provided via GitHub, supporting transparency.
Self-Containment: The manuscript is self-contained, presenting all necessary context for understanding the proposed methodology and results. The hypotheses, methods, and results are sufficiently detailed.

Suggested Improvements:
- Expand the introduction with more justification for using enhanced Paillier encryption over alternatives like differential privacy or blockchain-based schemes.
- Include a visual summary (e.g., flowchart) of the proposed FL-FedDT framework for better conceptual clarity.

Experimental design

Originality and Scope: The study presents original primary research relevant to the journal's aims and scope. It addresses a critical gap in privacy-preserving methods for federated learning in digital twin applications.
Research Questions: The research question is well-defined, and its significance is substantiated with a clear explanation of the knowledge gap.
Methodology: Methods are described in detail, ensuring replicability. The adaptive scheme employing the enhanced Paillier algorithm is robustly explained, and experimental configurations are appropriately detailed (e.g., simulation parameters, training model, and datasets).
Ethics and Standards: The study adheres to high ethical and technical standards.

Suggested Improvements:
- Clarify the rationale for key parameter selection, such as the choice of hyperparameters for the Paillier algorithm.
- Include a broader comparison with more baseline methods, such as state-of-the-art approaches in secure federated learning.

Validity of the findings

Data and Statistical Soundness: The data provided is robust and statistically sound. Experiments with different key lengths, training epochs, and configurations demonstrate the effectiveness of the proposed enhancements.
Conclusions: The conclusions are well-supported by the results. The study shows that the enhanced Paillier algorithm improves latency without compromising accuracy, addressing the trade-offs between security and computational efficiency.

Suggested Improvements:
- Further discuss the scalability of the proposed scheme in larger networks and more heterogeneous environments.
- Provide more in-depth analysis of the trade-offs between latency and security, particularly in real-world scenarios such as healthcare or industrial IoT.

Additional comments

- The manuscript is a significant contribution to the field of secure federated learning. It balances technical rigor with practical application, making it suitable for publication after addressing the minor revisions suggested above.
- The results are well-presented, but additional insights into practical deployment challenges (e.g., computational overhead on resource-constrained devices) would strengthen the manuscript.

Reviewer 2 ·

Basic reporting

The authors propose a trusted FL global aggregation method by using DTs.

Experimental design

To evaluate the eûciency of the proposed scheme, the authors conducted extensive experiments, with results validating that the proposed approach achieves training accuracy and security on par with baseline methods, while substantially reducing FL iteration time.

Validity of the findings

N/A

Additional comments

1. For the background in introduction, some recent works on DTs should be included.

2. The motivation hebind by using homomorphic encryption for privacy protection in FL needs to be well elaborated.

3. The authors are suggested to elaborate on how to achieve a high-fidelity digital twin in FL. For example, is there a need for real-time twinning pipeline in FL scenario; then what are the system and network requirements of implementing a real-time twinning?

4. How to determine the estimated deviations during the DT modeling?

5. How to indicate the effect of DTs on the FL process?

·

Basic reporting

While the paper presents a well-structured approach to privacy-preserving federated learning (FL) for digital twins (DTs), there are areas where further refinement can strengthen its scientific contribution. The following suggestions focus on improving the problem definition, mathematical clarity, experimental validation, and discussion of security implications.

1) Clarify the Research Gap in the Introduction
The introduction effectively presents the problem but does not clearly define the specific limitations of existing privacy-preserving federated learning methods. Expanding on what aspects of current techniques fail in digital twin (DT) applications will strengthen the motivation for this work. (Section 1: Introduction)

2) Improve Literature Review with a Comparative Analysis
While the related work section references various privacy-preserving techniques, it does not provide a direct comparison between the proposed enhanced Paillier encryption and alternative methods like secure multi-party computation (SMPC) or differential privacy. Including a comparative table or discussion would help readers understand its advantages and trade-offs. (Section 2: System Model and Preliminaries)

3) Refine the Mathematical Notation for Clarity
Some mathematical formulations, such as Equation (6) for global aggregation, introduce complex expressions without a clear step-by-step explanation. Providing a more intuitive breakdown of how the trust-based weight assignment is computed would improve clarity for a broader audience. (Section 3: Adaptive FL Scheme Utilizing an Enhanced Paillier Encryption for Privacy Protection)

4) Explain the Practical Impact of the Enhanced Paillier Encryption
The enhanced encryption method is stated to reduce computational overhead, but there is no clear quantification of how much this impacts real-world latency in a federated learning scenario. A more detailed performance breakdown comparing encryption times at different key sizes and their impact on training rounds would be beneficial. (Section 4: Performance Analysis)

5) Clarify the Trust-Based Aggregation Strategy
The trust-based model aggregation assigns weights based on learning quality and mapping bias, but how these weights are computed and updated over time is not clearly explained. A visual representation (e.g., a flowchart) of the process would aid in understanding. (Section 3.4: Trust-Based Aggregation in FL)

6) Address Potential Attack Scenarios in Security Analysis
The paper states that homomorphic encryption prevents plaintext inference attacks, but it does not address whether the system is resilient to poisoning attacks or gradient inversion attacks. Discussing how the method defends against these specific threats would strengthen the security claims. (Section 4.1: Security and Privacy Performance Analysis)

7) Improve Figure Annotations and Labels
While the figures are relevant, some (such as Figure 3 on encryption time) lack detailed axis labels and proper captions explaining the trends observed. Including error bars or standard deviation measures in performance plots would enhance the credibility of results. (Section 5: Experimental Evaluation)

8) Justify the Dataset Choice for Federated Learning Simulation
The use of MNIST for evaluation is standard but does not fully reflect heterogeneous digital twin environments. A discussion on why this dataset is appropriate for privacy-preserving federated learning in DT applications and how it generalizes to real-world use cases would be useful. (Section 5.1: Experimental Setup)

9) Expand Discussion on Communication Overhead
The study evaluates training accuracy and encryption efficiency, but there is limited discussion on the impact of increased communication overhead due to Paillier encryption. A section comparing communication efficiency with differential privacy or other cryptographic methods would strengthen the analysis. (Section 4.3: Latency Performance Analysis)

10) Provide a More Structured Conclusion with Future Work
The conclusion summarizes the contributions well, but explicit next steps—such as applying the method to real-world DT applications or optimizing encryption further—would make the research direction clearer. Suggesting future work on scalability to larger models or multi-party learning scenarios would be valuable. (Section 6: Conclusions and Future Work)

Experimental design

no comment

Validity of the findings

no comment

Additional comments

no comment

---

## Round 0.2 · Minor Revisions

Some minor comments are suggested before acceptance.

·

Basic reporting

While the literature cited is appropriate, the manuscript would benefit from brief engagement with more recent works on scalable and privacy-preserving FL in edge computing contexts to better situate the contribution within the current research landscape.

Experimental design

The experimental design is well presented, but an important detail missing is the specific procedure used to simulate non-IID data distribution across clients. The paper mentions non-IID settings but does not describe how the data heterogeneity was introduced. Clarifying this point is important for ensuring the experiments can be independently replicated.

Validity of the findings

The findings are mostly valid, but some additional rigor in the presentation of results would strengthen the paper. The experimental comparisons lack statistical measures such as standard deviation or confidence intervals, making it difficult to assess the consistency or variability of outcomes. Including multiple runs and reporting variance would support the reliability of the conclusions. Moreover, the criteria used in the trust evaluation framework are not quantitatively defined. Specifying how trust is computed and what thresholds or weighting mechanisms are used during aggregation would improve the clarity.

Reviewer 2 ·

Basic reporting

N/A

Experimental design

N/A

Validity of the findings

N/A

Additional comments

The authors have addressed all my previous comments. I have no further comments.

---

## Round 0.3 · accepted · Accept

Congratulations! Your paper has been accepted for publication.